# Evaluating the Structural Robustness of Large-Scale Emerging Industry with Blurring Boundaries

**DOI:** 10.3390/e24121773

**Published:** 2022-12-05

**Authors:** Yang Li, Huajiao Li, Sui Guo, Yanxin Liu

**Affiliations:** 1School of Economics and Management, China University of Geosciences, Beijing 100083, China; 2School of Business, Jiangsu Normal University, Xuzhou 221116, China; 3School of Management and Engineering, Capital University of Economics and Business, Beijing 100070, China

**Keywords:** percolation process, structural robustness, complex network, emerging industry

## Abstract

The present large-scale emerging industry evolves into a form of an open system with blurring boundaries. However, when complex structures with numerous nodes and connections encounter an open system with blurring boundaries, it becomes much more challenging to effectively depict the structure of an emerging industry, which is the precondition for robustness evaluation. Therefore, this study proposes a novel framework based on a data-driven percolation process and complex network theory to depict the network skeleton and thus evaluate the structural robustness of large-scale emerging industries. The empirical data we used are actual firm-level transaction data in the Chinese new energy vehicle industry in 2019, 2020, and 2021. We applied our method to explore the transformation of structural robustness in the Chinese new energy vehicle industry in pre-COVID (2019), under-COVID (2020), and post-COVID (2021) eras. We unveil that the Chinese new energy vehicle industry became more robust against random attacks in the post-COVID era than in pre-COVID.

## 1. Introduction

With the continuous innovation of technologies and the emergence of new consumer needs, emerging industries are gradually formed by providing new products or services. Generally, emerging industries represent the potential direction of industrial transformation that can effectively promote industrial structure upgrading and play an important role in long-term economic growth. Due to the significance of emerging industries, recent researchers have widely explored the evolution path and industrial structure of emerging industries [1,2], among which structural robustness has aroused wide attention from practitioners and scholars [3]. Structural robustness, in the context of industry, can be defined as the ability of the system to cope with disruptions [4]. As emerging industries are in their infancy, they are deemed to be more susceptible to external shocks and disruptions such as COVID-19, natural disasters, and financial crises. Thus, evaluating the structural robustness of emerging industries is helpful in understanding the emerging industries’ ability to battle disruptions and designing a robust industrial structure.

Presently, most emerging industries have evolved into complex ecosystems that include numerous interconnected firms and are highly intertwined with various mature industries [5]. Due to the interconnected structure, it is increasingly accepted that a network perspective is a significant way to investigate the structural robustness of emerging industries from a system scope [6].

In addition to the complexity of an interconnected structure, another significant feature of the emerging industry, compared to mature traditional industry, is the blurring of industrial boundaries, which is induced by the industrial fusion at the industry level and cross-industry conglomerates at the firm level [7]. It is difficult to claim whether a specific firm absolutely belongs to the target industry or not, which makes it difficult to effectively depict the network structure. To elaborate on the characteristics of blurring boundaries, we take the new energy vehicle (NEV) industry in China, for instance. From the perspective of industrial fusion, the NEV industry, as a strategic emerging industry in China, is fused with the traditional vehicle manufacturing industry, electrical power component industry, and the software industry of intelligent driving. From the perspective of conglomerates, NIO Co., Ltd, is an important NEV manufacturer and plays an irreplaceable role in the NEV industry. At the same time, NIO is also an important supplier of distributed power storage systems that is highly intertwined with the power storage industry. Moreover, the traditional Internet company, Baidu, also made efforts to develop NEVs in terms of its leading position in intelligent driving algorithms. In summary, industrial fusion and cross-industry conglomerates make emerging industries evolve into open systems with blurring boundaries.

Although effective boundary definition is the essential precondition to conducting industrial structure analysis, when complex structures with numerous nodes and connections encounter an open system with blurring boundaries, it becomes much more challenging to depict the structures of an emerging industry using current methods of network construction. Some research explicitly defines the industry boundary using firm-level business scope statements in their financial reports and analyst reports [8]. This investigation-based approach may be appropriate for simple small-scale emerging industries or mature traditional industries, rather than complex large-scale emerging industries. This is because, on the one hand, it is impractical to investigate the business scopes of all related firms for large-scale industries. On the other hand, the concept of industry structure is usually premature and unstable for emerging industries. The other research models the structure of emerging industries based on the datasets they used without explicitly defining the industrial boundaries [9]. This data-driven way usually underestimates the necessity of industrial boundary defining, which potentially affects the reliability and representativeness of their models. To sum up, although recent researchers emphasized that effective depiction of industrial structure is the critical element to investigating the evolution mechanism of a specific industry, most related research neglected the characteristic of blurring boundaries, especially when mapping the structure of a large-scale emerging industry.

In order to deal with the blurring boundary challenge when mapping the network structure of an emerging industry to evaluate industrial robustness, we propose a novel framework to map the network structure of a large-scale emerging industry, combined with the data-driven method and percolation theory. Inspired by the idea of using percolation theory to investigate the robustness of network structure (or more specifically, the abrupt appearance of the largest connected component) [10], we proposed a novel application of percolation theory to evaluate the completeness of industrial structure and thus deal with the blurring boundary challenge when mapping the structure of large-scale emerging industries.

The rest of this paper is organized as follows. First, we introduce the relevant literature and conclude the research gap in Section 2. Second, we describe the data used and propose our novel method for depicting the network structure of an emerging industry in Section 3. Further, we conduct network construction using the firm-level large-scale transaction data of the Chinese new energy vehicle industry from 2018 to 2021 and analyze the structural robustness against random and targeted attacks in Section 4. Finally, we conduct the experiment of robustness check and discuss the limitation of our method in Section 5.

## 2. Literature Review

### 2.1. Network Robustness

Robustness, in the context of networks, is interpreted as the ability of the network to withstand disruptions and maintain its major functionalities [11,12]. Related research has emphasized the importance of understanding network robustness and maintaining its functioning [13]. First, robustness is strongly correlated with network structure. Kim et al. [14] compared a set of network structures of four types of networks and their influences on robustness and indicated that network structure strongly affects network robustness. Their simulated results showed that a scale-free network is more robust. Li et al. [15] adopted network structural characteristics to depict supply chain robustness and demonstrated the advantage of using network structural characteristics over network types in terms of investigating network robustness. In addition, network robustness also depends on the types of attacks. A network tends to present different robustness in the face of random attacks and targeted attacks. Cats and Krishnakumari [16] explored whether networks with different structures and development patterns display different robustness characteristics in the face of random and targeted attacks. Their results showed that a multi-centric network is less robust than a uni-centric network.

In sum, as robustness and network structure are highly correlated, it is vital to have the network finely modelled and the structure accurately captured to measure robustness, especially for real-world network systems. As real-world networks are formed with complex interactions among actors, it is necessary to identify the critical components and characteristics when constructing a network. Our study focuses on the structural robustness of emerging industries having blurring boundaries. Different from traditional industries having mature industry structures, emerging industries are in their infancy. The industry structure and interactions among firms are always evolving, which makes it a challenge to identify and capture the critical components of the network for emerging industries. Based on the features of emerging industries stated above, we propose a novel method to select, identify, and construct the network structure of emerging industries. We applied our method to the case of the Chinese new energy vehicle industry and evaluated the structural robustness of the industry. Our study contributes to the literature regarding network construction of emerging industries.

### 2.2. Percolation Theory

Percolation theory is first considered as a gelation problem in the context of macromolecules [17]. After the development of the theory of critical phenomena, percolation became a widely used model to investigate the phase transition by modelling the percolation process in a lattice and computing different statistical and geometrical properties [18]. Two aspects of percolation lead to its impact. First, without too many preliminaries of dynamics or thermodynamics quantities, percolation is considered one of the simplest models that display a phase transition Second, percolation is sufficiently flexible in its definition to be matched with various problems, such as the dielectric response of in-homogeneous materials [19], epidemiology [20], or flows in porous media [21], among others [22,23].

With the rise of modern network theory, there is a renewed interest in percolation [24]. A network, broadly termed, is a set of nodes with arbitrary connections among them, compared to lattices, which are regular structures embedded in spaces of limited dimensions, with all the nodes having the same number of connections, i.e., the same degree for all the nodes. In this context, the removed nodes are usually deemed to be failures or attacks, and the largest connected component after the disturbances is interpreted to be the part of the network that remains operational. Specifically, the portion of a network that maintains major functions of the original network is seemingly the critical component of the whole network. Therefore, percolation has led to a deeper understanding of the robustness and resilience of real-world network systems [25,26,27], as well as providing new analytical tools [28] and appealing phenomenology from the perspective of statistical physics [29,30]. In sum, percolation is a widely used method to display the process of phase transition and performs promising applications in real-world network systems.

### 2.3. Emerging Industry

Emerging industries have been described as newly formed or re-formed industries that have been created by technological innovation, shifts in relative cost relationships, the emergence of new consumer needs, or other economic and sociological changes that elevate a new product or service to the level of a potentially viable business opportunity [31]. Disruptive innovation is the essential driving force of industrial emerging and leads to some very different value propositions for products and services. In addition to technological driving, Srai et al. [1] explored the characteristics of the emerging industries using cross-case analysis and identified several other common characteristics: intertwining with traditional industries, many embryonic firms and spin-offs, and uninformed supplier–buyer relations.

Emerging industries have been widely discussed in detail. Many related studies focused on the formation and evolution mechanism of emerging industries. Jacobides [32], Funk [33] undertook the retrospective analyses of representative industries. Cao [34] adopted the SWOT analysis to discuss the superiority of emerging industries in China. Awa [35] predicted at which stage of the maturity curve a new product will be suitable and tailored to the product development process. These studies are too conceptual to reveal informative details and insights on evolution characteristics of the industry network. Furthermore, with the development of big data technology, the barrier to acquiring industrial data dampens. Therefore, the role of industry structure analysis based on real-world data has been gradually important to understand the development of emerging industries. Wang et al. [36] explored the dependence structure of the 5G (the fifth-generation mobile communication technology) industry in China and found that the dependence structures of different firms are heterogeneous, and most firms are strongly interconnected with their partners in the 5G industry. Srai et al. [1] investigated the solar photovoltaic industry by analyzing its booming supply network and demonstrated a positioning shift of the critical firms in the industry. As industrial structure analysis based on real-world data can provide abundant information from a system perspective, and network analysis has become a trendy method to understand the intrinsic mechanism of industrial emergence.

### 2.4. Mapping the Network Structure of Industry Systems

With the increasing complexity of modern industries, network analysis has become a key instrument to demonstrate the static and dynamic characteristics of industry structure. Li and Pei [37] constructed an improved local world model based on the innovation network to explore the evolution of emerging industries. It was found that the node degree follows the power-law distribution. Through the unguided SBM and network DEA model, Zhong et al. [38] analyzed the technology innovation efficiency of strategic emerging industries in China at the province level from 2002 to 2013 and found that during this period the overall technology innovation efficiency tended to continuously improve, and, for eastern provinces, this efficiency is even higher than that of central and western provinces. Mapping industrial structures into network models has been a popular way to explore industrial development.

In previous studies on the mapping industrial network, three types of networks—innovation-based network, correlation-based network, and physic-based network—usually depict the structural characteristics of industries, providing effective information from different perspectives [39]. Innovation-based networks, such as research project cooperation [40] and joint patents [41], are applied to explore the technological landscape of emerging industries through an innovation lens. Correlation-based networks are based on financial information to capture the implicit structure across a specific industry, such as stock return [42], financial contagion [43], and risk spillover [44]. Physic-based networks are based on real-world data among firms in an industry, such as supply chain interaction [45], collaboration [46], and transaction interaction [39]. According to the reliability of physic-based networks, which usually represent actual interactions across the focal industry, physic-based networks based on real-world data can depict the industrial structure more precisely.

Although the anatomy of industry structure based on network mapping performs increasingly significant roles, most of the network mapping methods do not take the blurring of industrial boundaries, the significant characteristic of an emerging industry, into consideration, which potentially limits the application of network analysis when modelling the structure of an emerging industry.

### 2.5. Research Challenges

To sum up, on the one hand, a complex structure and blurring boundaries make it challenging to effectively map the network structure of an emerging industry. On the other hand, percolation has brought a deeper understanding of real-world complex systems. Therefore, this study proposes a novel framework to effectively depict the structure of the emerging industry with the consideration of the complex structure and blurring boundaries and demonstrated the potential of using the percolation process for mapping the economic system.

## 3. Data and Methodology

### 3.1. Data Description

In this study, we select the new energy vehicle industry in China to conduct the empirical experiment. We chose the transaction data of China’s NEV industry from the ChinaScope database (https://inews.chinascope.com, accessed on 12 August 2021) to map network structure based on capital flow. The ChinaScope database collects information from publicly disclosed financial reports, prospectuses, and reports from financial analysts that is standardized in the manner of supply chains.

### 3.2. Methodology

To extract the main structure of the emerging industry, we first select related listed firms as the major component of the emerging industry based on industry investigation. Second, taking selected firms as initial clusters, we recursively crawl the transaction data based on breadth-first search (BFS) until generating the percolation cluster, which means the appearance of a giant connected component. Finally, we use the random walk to remove less related firms and quantify the inter-connectivity among listed firms. The diagram of the method is shown in Figure 1.

#### 3.2.1. Step 1: Initial Cluster Definition

We first organize the basic structure of the NEV industry according to the report in ChinaScope. The starting point of our method is based on the basic concept of NEV industry structure, which reveals a simple linear structure. Because of industrial fusion and cross-industry conglomerates, the real-world industry structure is complex and non-linear, as shown in Figure 2. Then, according to Wang et al. [36], as the major participants, the listed firms are used to represent the main units of the target industry, such that we can capture the inter-connected structure among listed firms to effectively represent the structure of the target industry. Thus, we select the listed firms whose business scope is involved in the basic component of the NEV industry. As shown in step 1 of Figure 1, big blues nodes represent listed firms whose business is involved in the NEV scope. Small grey nodes represent suppliers or customers of listed firms. In the context of percolation, these listed firms and their trade partners are abstracted as initial clusters in the unknown high-dimension lattice, which represents all transaction information among firms. The lattice is unknown because it is inaccessible to all transaction information of the target industry.

#### 3.2.2. Step 2: Percolation Cluster Generation

The aim of this step is to acquire enough transaction data to describe the structure of the target industry. The core of this step is that, to what extent, so-called ‘enough’ transaction data are reached. Our basic idea is that the transaction information is fundamentally enough to depict the structure of the target industry when the entire network is interconnected.

In the last step, we acquire the initial clusters using the direct trade information of listed firms. Although the present industries are generally interconnected as an entirety to produce new services or products that are widely mentioned in the literature (Wang et al. [36], Osadchiy et al. [47], Liu et al. [48]), most initial clusters are disconnected from each other. This is because we lack enough transaction information to depict the overall landscape of the target industry. Hence, our main assumption is that listed firms whose business scope is involved in the NEV industry (as shown in Figure 2) are modelled as the major units of the target industry and they are supposed to be interconnected as an entirety. Thus, starting with the initial cluster, we continue to collect transaction information of related firms using breadth-first search. It means that we continue to collect transaction data of listed firms’ trading partners. These added edges are represented by the orange edges in step 2 of Figure 1. As more transaction data are added, more transaction edges are added to the initial clusters and thus initial clusters are growing. When the entire network is connected, our transaction information is fundamentally enough to represent the target industry in its entirety.

In the context of percolation, the initial clusters can be seen as the boundary of a high-dimension lattice. All real-world transaction data can be seen as this lattice. The data crawling process can be seen as the process of percolation cluster generation. If the number of connections (*l*) increases at $l_{c}$, these initial clusters grow and coalesce, leading to the emergence of a large cluster. When all initial clusters are interconnected as a whole, all nodes in the lattice belong to the same cluster. The entire cluster is the so-called percolation cluster as it reaches the boundaries of the lattice. To quantify the process of this phase transition, we use order parameter *P*, which represents the probability that a randomly chosen node belongs to the largest cluster.

The above process is a so-called data-driven percolation process. We show the process in Figure 3. The big blue nodes represent listed firms. The small grey nodes represent non-listed firms. The arrows represent the transaction links. According to the BFS, the first layer transaction links are the transactions between listed firms and their first-tier suppliers/customers. The second layer transaction links are the transactions between the first-tier suppliers/customers of the listed firms and the second-tier suppliers/customers of the listed firms. We continue to crawl the transaction information based on BFS until the major units are interconnected with each other.
(1)$$P\left(n\right)=\frac{n_{k}}{n},$$
where *n* represents the overall number of nodes, and $n_{k}$ represents the number of nodes in the largest cluster. When P(n) gets close to 1, the percolation cluster emerges, revealing that all nodes belong to the same cluster. Because *P* represents the percentage of the largest cluster to the entire network, we then use *P* to quantify the scale of the giant cluster (which is shown in Section 4.1.

#### 3.2.3. Step 3: Pruning Based on Random Walk

The aim of this step is to abstract the major structure of the target industry. In the last step (step 2), we acquired enough transaction information to depict the interconnected structure. However, we take a large number of firms that are less related to the target industry into consideration at the same time. To deal with this problem, we can calculate the connected structure among major units rather than determine whether every node or link belongs to the target industry. Hence, listed firms are represented as nodes. The interconnections among these listed firms are represented as edges and quantified based on the random walk process. In other words, firms that are among the listed firms are kept. Firms that are not among the listed firms are removed. In order to clarify the process, a diagram is shown in Figure 4.

More specifically, it is impractical to clarify whether numerous embryonic firms and spin-offs belong to the target industry. In order to reduce the redundancy, we simulate the random walk process on the percolation cluster and only quantify the interconnection among the listed firms. In other words, we only take the related firms among listed firms into account and abstract these related firms into edges instead of nodes. Through the interconnected structure among these main units (listed firms), we not only keep the major structure of the target industry but also cut off the less related nodes so that we can deal with the blurring boundary challenge.

We quantify the weighted degree of the inter-connected relation from both the demand perspective and supply perspective. The capital flow is represented by $x_{ij}$. The direction of demand flow is the same as the direction of capital flow, whereas the direction of supply is the opposite. When calculating the demand flow, we first set the initial demand state as:(2)$$D_{0}^{i}=\left(d_{1}^{0},d_{2}^{0},d_{3}^{0},\cdots ,d_{i}^{0},\cdots ,d_{n}^{0}\right),$$
where $d_{i}^{0}=1$ and others equal 0. Then the demand transition matrix can be represented as:(3)$$DT=dt_{ij}=\frac{x_{ij}}{\sum_{i}x_{ij}}.$$

After the *K*-order transition, the demand state equals:(4)$$D_{K}^{i}=\sum_{k=1}^{K}D_{0}^{i}\times DT^{k}=\left(d_{1}^{K},d_{2}^{K},d_{3}^{K},\cdots ,d_{i}^{K},\cdots ,d_{n}^{K}\right).$$

Thus, the demand flow from firm *i* to firm *j* equals:(5)$$DC=dc_{ij}=\frac{d_{j}^{K}}{K},$$
where $dc_{ij}$ represents the proportion of initial demand of node *i* on node *j* after the *K*-order transition.

Similarly, when calculating the supply flow, we first set the initial supply state as:(6)$$S_{0}^{i}=\left(s_{1}^{0},s_{2}^{0},s_{3}^{0},\cdots ,s_{i}^{0},\cdots ,s_{n}^{0}\right),$$
where $s_{i}^{0}=1$ and others equal 0. Then the supply transition matrix can be represented as:(7)$$ST=st_{ij}=\frac{x_{ji}}{\sum_{i}x_{ji}}.$$

After the *K*-order transition, the supply state equals:(8)$$S_{K}^{i}=\sum_{k=1}^{K}S_{0}^{i}\times ST^{k}=\left(s_{1}^{K},s_{2}^{K},s_{3}^{K},\cdots ,s_{i}^{K},\cdots ,s_{n}^{K}\right).$$

Thus, the supply flow from firm *i* to firm *j* equals:(9)$$SC=sc_{ij}=\frac{s_{j}^{K}}{K},$$
where $sc_{ij}$ represents the proportion of the initial supply of node *i* on node *j* after the *K*-order transition.

Finally, we sum up demand flow and supply flow between node *i* and node *j* to represent the overall inter-connected relation among listed firms:(10)$$R=r_{ij}=SC_{ij}+DC_{ij}.$$

A higher inter-connected relation from node *i* to node *j* means a larger proportion of firm *i*’s supply and demand flowing into firm *j*.

As we can acquire listed firms’ business scope according to their financial reports, we can make sure these listed firms are involved in the NEV industry. However, we cannot know the business scope of other firms. In our model, these non-listed firms (NEV-related firms as shown in Figure 5) function as the connections instead of nodes. The listed firms can be seen as the blurring boundary of the target industry. Therefore, the listed firms belonging to the industry that is shown in Figure 2 are modelled as nodes. The other firms are either considered as connections or removed.

Based on the aforementioned framework, we can depict the inter-connected structure of the target industry To sum up, taking listed firms whose business scope is involved in the target industry, we implicitly define the boundary of the target industry. Through percolation cluster generation based on BFS, we explore the necessary inter-connected structure of the target industry. Through pruning based on a random walk, we ignored the less related firms such that we can rationally depict the industry structure with blurring boundary challenges.

#### 3.2.4. Robustness to Random Attacks and Targeted Attacks

In our study, we apply the simulation method to explore the robustness under random and targeted attacks. The types of attacks can be typically classified as random attacks and targeted attacks. The random attack is able to act on any firms and interconnections in the industry network, and the targeted attack follows certain rules when exerting impact. Specifically, for random attacks, firms are given an identical likelihood to be removed away from the network. Once a node is removed, all the interconnections tied to it are removed as well. For targeted attacks, firms are sorted and removed according to their strength, which is denoted by the sum of all linked weights of neighbours of the focal node. Then, the simulation will carry on to remove the firms connected to the attacked firms. Eventually, the average network efficiency is calculated as the measurement of robustness. In addition, as the random attack is highly uncertain in terms of the first removed firm, we apply the Monte Carlo simulation 1000 times and take the average of the results.

The average network efficiency is used to depict the ability of information transmission between nodes in the network. By analyzing the process of network efficiency loss when more nodes are removed, we can capture the robustness of the industry network. The average network efficiency is defined as $E(G_{\alpha })$:(11)$$E\left(G_{\alpha }\right)=\frac{1}{N(N-1)}\sum_{i\neq j}e_{ij}\left(G_{\alpha }\right)$$
where α represents the percentage of removed nodes, *N* represents the total number of firms in the industry network *G*, and $e_{ij}\left(G_{\alpha }\right)$ represents the direct interconnections between firm *i* and *j*.

Furthermore, we use the area under the curve (AUC) value of network efficiency as a supplementary measurement to explore the robustness of the industry network by integrating the area under the network efficiency curve. The formula is defined below:(12)$$AUC_{\alpha }=\int_{0}^{\alpha }E\left(G_{\alpha }\right)$$
where α is the percentage of removed firms.

## 4. Empirical Experiments and Results

Our study proposes a novel framework to deal with the blurring boundary challenge when mapping the structure of large-scale emerging industries. In the above section, we introduced the method to map the structure of a large-scale emerging industry. In this section, we apply the method to the NEV industry and construct the industry network. In addition, we explore the differences in network characteristics between industry networks constructed with different volumes of transaction data (from 1-layer to 5-layer supplier–buyer relationships). By comparing the overlapping components between these industry networks, we demonstrate that, in the case of the NEV industry, acquiring three layers of transaction data are sufficient to map the major structure of the entire network. Then, based on the industry network constructed with 3-layer supplier–buyer transaction data, we analyze the robustness of the industry network under random attacks and targeted attacks. By comparing the evolution of network efficiency in the face of different patterns of attacks, we can capture the structural robustness of a large-scale emerging industry, i.e., the NEV industry in China.

### 4.1. Network Construction

The industry network constructed through our method consists of 139 listed companies (as nodes) and their interconnections (as edges). In addition, we construct a group of networks that are based on different layers of supplier–buyer relationships. Then, we compare the giant percentage and number of nodes among these networks and present the results in Figure 6. The results show that with the increase in layers of supplier–buyer relationships, the giant percentage increases quickly initially to around 90% until the layer reaches 3. After that, the giant percentage stays at the level of 90%. As for the number of firms, it exhibits a continuously rising trend with the increase in layers of supplier–buyer relationships. When incorporating more layers of supplier–buyer relationships into the network, of course, the number of firms involved becomes larger and larger. However, although more nodes are added to the network, the giant percentage does not have a corresponding addition. It indicates that those newly added nodes do not have close connections with the giant connected component or are peripheral firms. Thus, considering too many nodes when mapping the network is not efficient, because no advancement is found in the giant connected component but more noises are included.

To further examine the above argument, we also compare the overlapping rate of edges between these networks. The results are shown in Table 1. The overlapping rate is extremely small between the networks of 1- and 2-layer supplier–buyer relationships. Then, we find a huge increase in the overlapping rate when we compare the networks of 2- and 3-layer supplier–buyer relationships. It indicates that the networks of 2- and 3-layer supplier–buyer relationships are alike. The results indicate that when the layer of supplier–buyer relationships reaches 3, a further increase in the layer cannot add more effective information to the network. Therefore, in the following robustness analysis, we use the industry network of 3-layer supplier–buyer relationships.

### 4.2. Robustness Analysis

According to the comparison above, we find that 3-layer buyer–supplier transaction data are optimal to map the industry network. Thus, our robustness analysis is based on the industry network of 3-layer supplier–buyer relationships. We explore the structural robustness of the industry network in the face of random attacks and targeted attacks in 2019, 2020, and 2021. A simulation process is conducted to see the evolution of network efficiency, which reveals the ability of the network to maintain its major function and survive the attack. The results are presented in Figure 7.

First, when the industry network is under random attack, we find that the curves of network efficiency from 2019 to 2021 have the same shape. The network efficiency experiences a gradual reduction with the increase in failed nodes. In addition, as marked in the figure, when 20% of nodes are removed due to random attacks, the maintained network efficiency is 0.23, 0.2, and 0.28 in 2019, 2020, and 2021, respectively, exhibiting that the structural robustness of the industry network decreases from 2019 to 2020 and bounces back to an even higher level from 2020 to 2021. The condition is similar when we take the AUC value as a supplementary measurement. The AUC value of network efficiency is 0.097, 0.09, and 0.11 in 2019, 2020, and 2021, respectively, which also exhibits the pattern of initially falling and subsequently rising. When we assume that 2019 is the year before the COVID-19 outbreak (pre-COVID), 2020 the year under COVID-19, and 2021 the post-COVID year, we can see that the structural robustness of the NEV industry experienced a reduction under COVID-19, but it recovered later in 2021 and became even better than the pre-COVID year. It indicates that the NEV industry became more robust against random attacks in the post-COVID era.

Second, when the industry network is under targeted attack, the process of network efficiency loss is completely different from that under random attack. Taking 2019 as an example, the network efficiency reduces gradually in the initial stage and experiences an abrupt drastic fall, losing 70% of the original value when the percentage of failed nodes approaches 10%. In 2020, the initial stage when the network efficiency gradually falls ends instantaneously and is followed by an abrupt drastic fall, losing about 80% of the original value. In the post-COVID year, the process of network efficiency loss is similar to that in the pre-COVID year, whereas the initial stage lasts longer. The difference is also significant when considering the AUC value. The AUC value is 0.072, 0.029, and 0.102 in 2019, 2020, and 2021, respectively. This indicates that the structural robustness of the NEV industry experienced a great reduction during the pandemic and an even greater recovery in 2021, suggesting that the NEV industry becomes more robust against targeted attacks in the post-COVID era. In addition, it also indicates that during the pandemic, the NEV industry was particularly vulnerable to targeted attacks, suggesting that we should also pay attention to potential targeted attacks rather than only focus on the impact of the pandemic.

Third, when we compare the performance of robustness between the situations of random and targeted attacks, we find that the NEV industry is more robust to random attacks than to targeted attacks both before and after the pandemic. In the pre-COVID year, the AUC value is 0.097 in the case of random attacks and 0.072 in the case of targeted attacks, i.e., 34.7% higher. In 2020, the AUC value in the case of random attacks only decreases by a small amount compared to 2019 (from 0.097 to 0.09), whereas in the case of targeted attacks, the AUC value loses more than half compared to 2019 (from 0.072 to 0.029). In the post-COVID year, the AUC value bounces back and surpasses the level of 2019, but the value in the case of random attacks remains higher than that in the case of targeted attacks, i.e., 0.11 and 0.102, respectively; despite that, the value in the case of targeted attacks recovers more. However, although the robustness bounces back and becomes even more robust in 2021 than in 2019 in both cases, the improvement in the case of targeted attacks is 40.7% from 0.079 pre-COVID to 0.102 post-COVID, whereas the improvement in the case of random attacks is only 13.4% from 0.097 pre-COVID to 0.11. In addition, although it seems that the robustness against targeted attacks has approached the level in the case of random attacks, we still argue that the robustness against random attacks is explicitly better due to the existence of the abrupt drastic fall in the case of targeted attacks. Specifically, the process of network efficiency loss in the case of random attacks is relatively stable and gradual, which provides industrial practitioners with effective predictions of the future and sufficient time to react, whereas in the case of targeted attacks, there is a potential abrupt drastic fall, which makes it harder to get effective predictions and feasible reactions. In addition, a stable process of reduction will give sufficient time for industrial participants to adjust their expectations and emotions to the future performance of the industry, but the abrupt dive will arouse panic in unprepared people, causing irrational reactions.

In sum, the above results indicate that (1) the structural robustness of the NEV industry is better in the face of random attacks than targeted attacks, (2) the industry became more robust against random attacks in the post-COVID era than in the pre-COVID era, (3) the robustness against targeted attacks was improved far more than that against random attacks, and (4) the existence of abrupt drastic fall in the case of targeted attacks may lead to uncertainty and externality when battling the attacks.

The NEV industry shows stronger robustness when facing random attacks. Previous research examined the robustness of random networks (ER networks) and scale-free networks (BA networks), which demonstrated that random networks are robust to random and targeted attacks and scale-free networks are only robust to random attacks but vulnerable to targeted attacks [49]. The reason may be that the scale-free network has a centralized structure with hub nodes having a large number of connections. In the targeted attacks, those hub firms are first removed, causing the network to lose plenty of interconnections instantaneously. In addition, in a scale-free network, to chase operative efficiency, a new firm is prone to develop relationships with hub firms, which makes a scale-free network become more power-law distributed [50]. The NEV industry is less robust to targeted attacks. The reason could be the existence of hub firms such as CATL, which is the dominating firm producing vehicle batteries.

The above discussion has revealed that the robustness against targeted attacks is highly correlated to the degree distribution of the industry network. Specifically, a centralized distribution indicates weak robustness against targeted attacks (namely, a negative correlation). Thus, we fit the degree distributions using weighted degree from 2019 to 2021 and compare the trend of degree distribution to that of robustness against targeted attacks. The results are presented in Figure 8. The functions in the figure represent the fitted lines, and the shaded areas represent the confidence interval at a 95% confidence level. The absolute value of the function’s gradient is called the power index, which indicates the extent of being power-law distributed. Because power-law distribution indicates centralization, the magnitude of the power index indicates the extent of centralization. As a result, the distribution presents a significant power-law characteristic (normally a number between 2 and 3 is considered to be a power-law distribution). From 2019 to 2020, the power index increased from 2.96 to 3.01, which means that the industry network became more centralized from pre-COVID to under-COVID. Then, from 2020 to 2021, the power index fell back to 2.2, which means that the degree distribution became less centralized in the post-COVID stage. In comparison, the robustness against targeted attacks dropped from 2019 to 2020 and then bounced back in the following stage, which is opposed to the trend of centralization (degree distribution). Thus, the trend of centralization (degree distribution) is negatively correlated to the trend of robustness against targeted attacks, which supports the discussion above.

The structural robustness in the post-COVID phase can be even higher than that in pre-COVID. It suggests that, despite the destruction, it is also a chance for industrial participants to reorganize their supply chain management strategies and facilitate innovations such as digital platforms and industry 4.0 to adapt to the pandemic [51,52]. For the NEV industry, intelligent and automatic production has been the trend for practitioners to chase, which has been a solid impression attached to the industry. It is reasonable to assume that the rebound of robustness benefits from those intelligent production strategies. Intelligent production can help NEV manufacturing firms quickly fit in the ‘new norms’ in the post-COVID era with less dependence on humans. That is why the NEV industry has higher robustness to targeted attacks in post-COVID than in pre-COVID.

## 5. Conclusions and Implications

The structural robustness of large-scale emerging industries is a topic with widespread concern. However, the phenomenon of blurring boundaries has made it a challenge to map the industry network for a large-scale emerging industry. To cope with this challenge and analyze the structural robustness of a large-scale emerging industry, our study proposes a novel framework of methodology based on industry investigation, data-driven methods, and percolation theory, and applies our method to the NEV industry, a typical large-scale emerging industry. Our results show that to capture the major part of the NEV industry and construct the industry network, 3-layer buyer–supplier transaction data are sufficient. Furthermore, we conduct simulation experiments to examine the structural robustness of the NEV industry against random attacks and targeted attacks. The results show two major findings: (1) the robustness to random attacks of the NEV industry is better than that to targeted attacks; (2) the industry became more robust against random attacks in the post-COVID phase. In addition, our results also show that the robustness against targeted attacks has a larger rebound to the same level as that against random attacks in the post-COVID phase. However, the existence of an abrupt drastic dive in the case of targeted attacks indicates that even though the static robustness levels are similar between the cases of random and targeted attacks, the process of network efficiency loss in the case of targeted attacks is unstable and even severely volatile.

Our results provide the following implications. First, the framework of the methodology proposed in our study is customized to map a large-scale emerging industry with blurring boundaries. We argue that to conduct network analysis, researchers should first pay attention to appropriately mapping the network structure. Then, there exists a question that how to deal with the blurring boundary issue of the present industry, especially for those emerging industries. Our work and the methodology presented seeks to recognize, filter, and determine the most related entities and the most critical and influential relationships, and eventually model the network structure of emerging industries. We suggest that, for policymakers, it is important to have an overall understanding of emerging industries’ network structures, which consist of the most important firms and relationships. Thus, there is no need to have the industry modelled with numerous small firms and less-related firms. In contrast, focusing on the most related and influential firms (namely listed firms) and inter-relationships can eliminate distractions and noise. Policymakers should review the network structure of emerging industries with blurring boundaries and refine the range of those industries. For industrial practitioners, they should also review their position in the industry, which can help them better understand the condition of current business and underlying directions for future development. We suggest that managers focus not only on their close partners and competitors but also on the influences of the whole industrial system. Risks and attacks could be from anywhere in the whole system. Our work can help them know better about the relationships with other large firms, which can rarely be observed directly. For academics, we suggest that, when considering network construction, one should also consider the blurring boundary issue and determine the list of related nodes to include in the network. Second, because the network of the NEV industry is a scale-free feature, making it hard for the network itself to build robustness against targeted attacks, policymakers should regulate the self-development of the NEV industry and lead the industry to be robust against targeted attacks. In addition, more attention should be paid to firms with large importance (namely node strength), as they are the first to be attacked in a targeted attack scenario. In addition, support should also be given to small and medium firms, as they serve as ties connecting large firms and spreading the influences of attacks. For industrial practitioners, they should be cautious of possible sources of targeted attacks rather than only pay attention to improving operative efficiency. Third, the process of network efficiency loss in the case of targeted attacks is unstable. It makes predictions and expectations of the future trend under targeted attacks ambiguous. Thus, the government and practitioners should conservatively or cautiously predict the development of targeted attacks’ impacts and always be prepared to encounter a sudden deterioration of the industry. In addition, as a sudden fall happens when those most critical firms are removed, monitoring the operation of those critical firms can provide warning signals for the sudden loss of the viability of the industry. In addition, supporting those critical firms to help them maintain their operations can prevent the occurrence of a sudden fall. Thus, to deal with targeted attacks, policymakers should pay attention to those ’targeted’ critical firms and provide corresponding policies and regulations to maintain their performance.

## Figures and Tables

**Figure 1 entropy-24-01773-f001:**
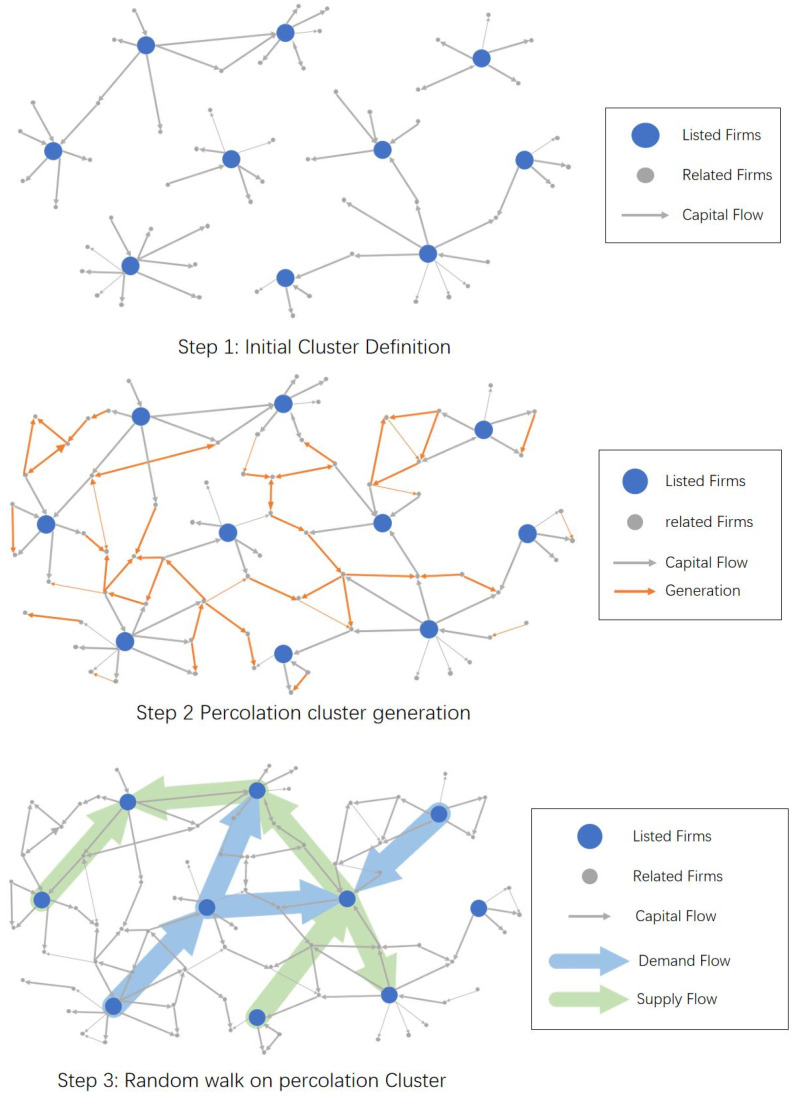
The diagram of the method.

**Figure 2 entropy-24-01773-f002:**
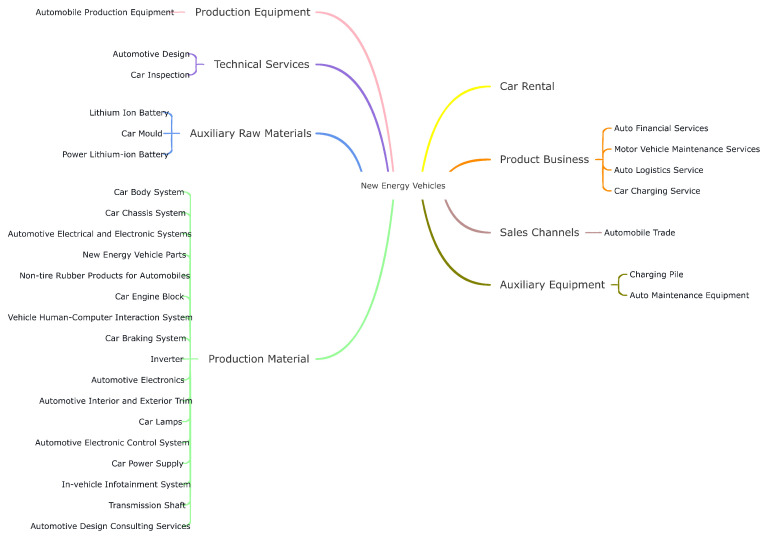
The sector composition of the NEV industry in China.

**Figure 3 entropy-24-01773-f003:**
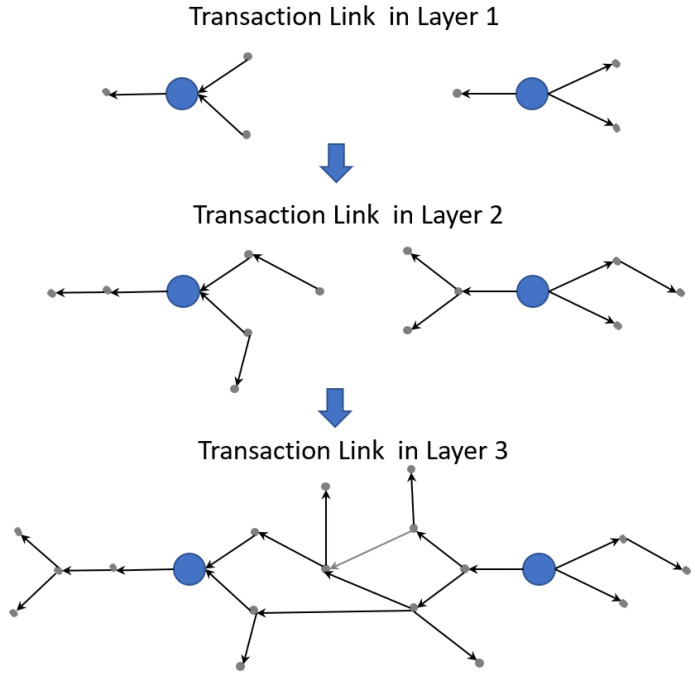
The diagram of percolation cluster generation.

**Figure 4 entropy-24-01773-f004:**
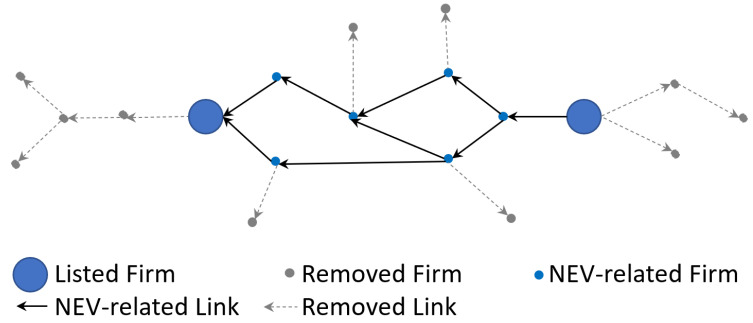
Pruning based on random walks.

**Figure 5 entropy-24-01773-f005:**
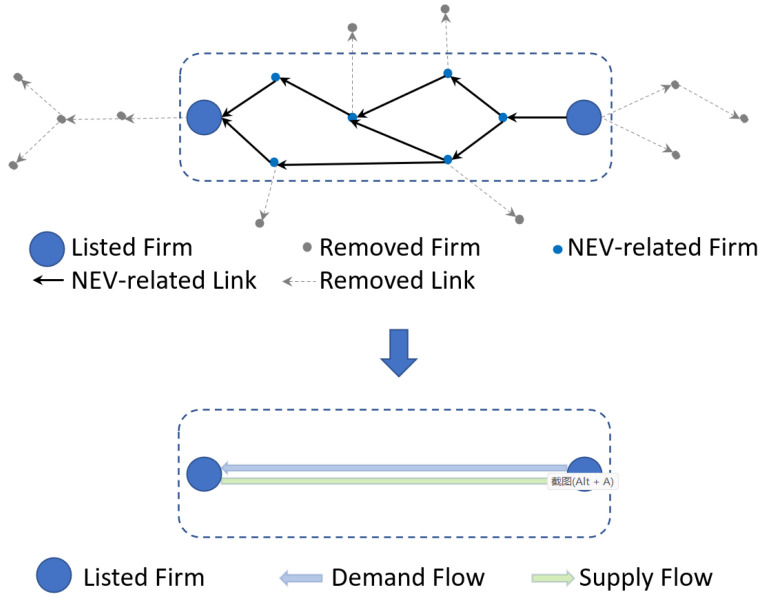
Depiction of the industry structure with blurring boundaries.

**Figure 6 entropy-24-01773-f006:**
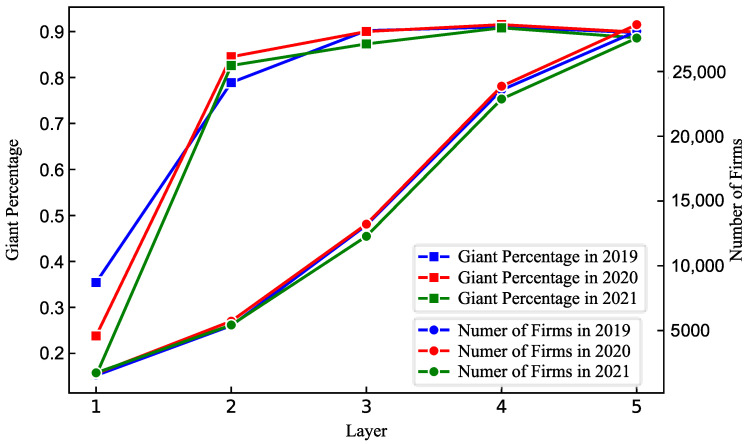
The comparison of giant percentage and number of nodes with different layers of transaction data. The giant percentage measures the proportion of the nodes in the giant connected component to the whole network constructed from 1-layer transaction data to 5-layer transaction data. The number of firms is the number of nodes involved in the network constructed with different layers of transaction data.

**Figure 7 entropy-24-01773-f007:**
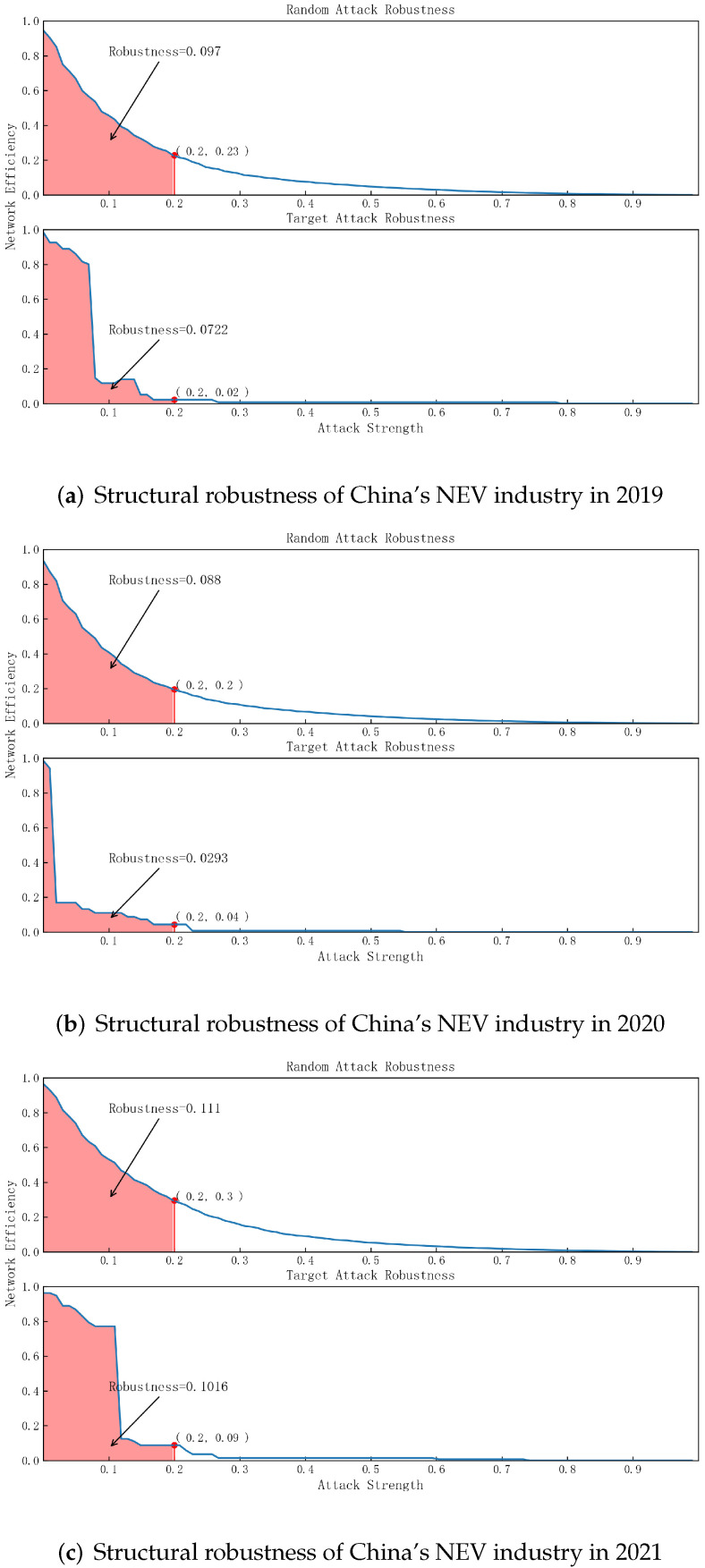
The simulated process of network efficiency in the face of random attacks and targeted attacks from 2019 to 2021. For each year, we present the simulated results under random attacks and targeted attacks. The x-axis is the percentage of failed nodes. The y-axis is the network efficiency. When the network is under attack, the proportion of failed nodes increases and the network efficiency decreases. The red area is the measurement of AUC, which is the integral of network efficiency when 20% of nodes are removed. A larger AUC value represents better robustness when controlling the percentage of failed nodes.

**Figure 8 entropy-24-01773-f008:**
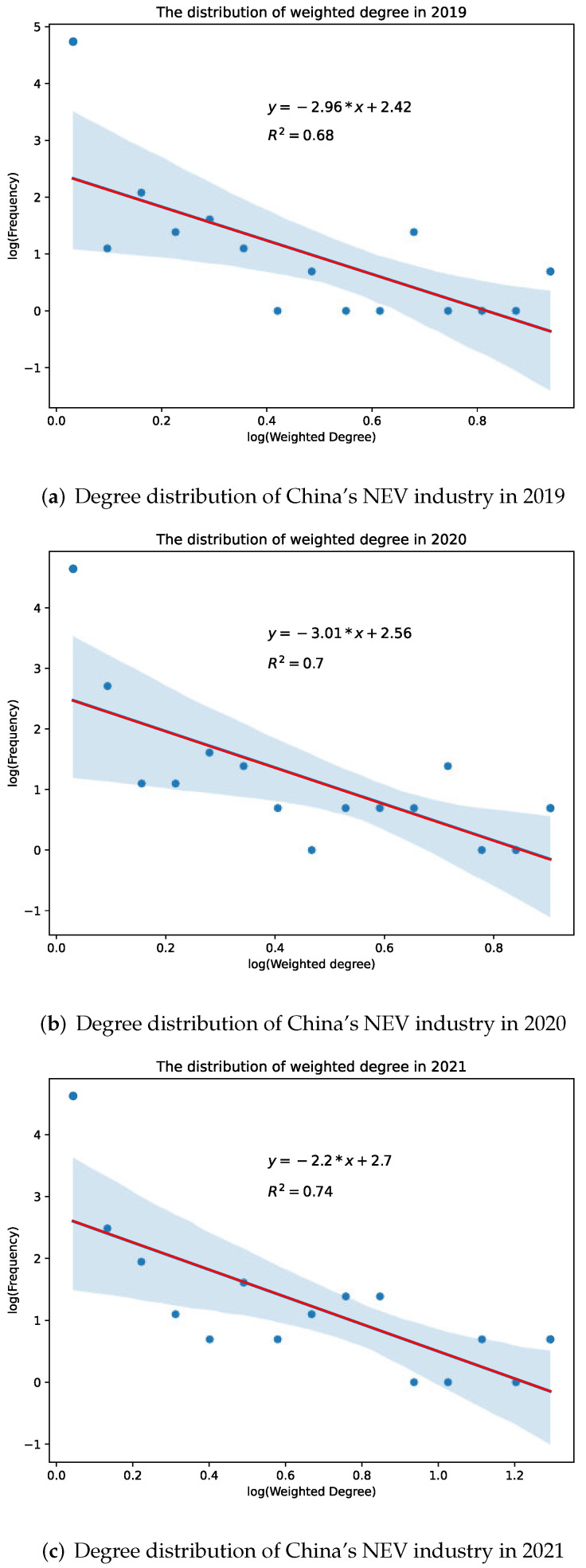
The degree distribution of China’s NEV industry from 2019 to 2021.

**Table 1 entropy-24-01773-t001:** The overlapping of edges between the networks constructed with different layers of transaction data.

Layer	2019	2020	2021
1-2	0.45%	0.38%	0.15%
2-3	85.67%	90.32%	68.08%
3-4	100.00%	91.46%	91.43%
4-5	100.00%	100.00%	95.98%

## Data Availability

The data that support the findings of this study are available from the corresponding author upon reasonable request.

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
