# Peer review of "Evaluating the Structural Robustness of Large-Scale Emerging Industry with Blurring Boundaries"

_entropy, 2022, doi:10.3390/e24121773_

Round 1
Reviewer 1 Report
The manuscript provides a solid context for the subject matter and advances overall understanding of the field. The theoretical foundation is well-outlined and provides a substantive foundation for the authors' argument. The methodology is sound and aligns with appropriate rigor. The conclusions are well-articulated and provide an interesting narrative for the readers. However, the manuscript seems to end abruptly. I would encourage the authors to consider the intellectual merit (implications for the field) and broader impact (implications for society) and share these accordingly. The manuscript is strong and has considerable implications, but a stronger close to the manuscript that includes a more intentional implications section would add considerably to its overall value.
Reviewer 2 Report
The manuscript presents a robustness analysis for large-scale emerging industries with blurring boundaries, modelized as networks. The robustness analysis is performed as the classical dynamics of random and targeted attacks in complex networks.
The paper considers a very interesting problem but it needs some refinement to be accepted for publication. In the following, I list some major comments:
- As the first comment, I noted that the difference between random and targeted attacks is never described within the paper. Some definition is given in Section 3.2.2 but it misses describing the measure used in order to define the node's importance (degree? betweenness? Etc.).
- The mechanism of network construction of Step 2 in Section 3.2 is not clear. Since this step is the core of the network generation, it deserves a clearer description of how the links are added, the data used to infer their presence, the hypothesis, etc.
- The same situation for Step 3, the ratio used by the authors to increase the network's density is not completely clear. Moreover, in line 265, it is not clear under what hypothesis the interconnected structure is considered a major structure and how nodes are deleted.
- I have an issue with the definition of blurred boundary. In particular, it is never formally defined, thus is difficult to understand if such a boundary is composed of secondary industries (as it appears to be following my previous comment) or industries involved in a NEV. In this second case, many of such nodes could be potentially very big industries (hub nodes). In this vein, they could have a bigger role in the analysis of resilience – especially in the case of targeted attack simulations - but this aspect is never discussed.
- It is not clear how the framework is able to create an inter-connected structure (line 289) and an example could help the reader.
- In line 290 the manuscript refers to a network in Figure 4, but Figure 4 refers to the simulated process of network efficiency.
- Line 327, what do layers represents?
- Section 4. The manuscript lacks a clear description of the network generation. Moreover, in the introduction is claimed that such a framework could consider large networks, but as far as I can see, the network is composed of only 139 nodes. Finally, the robustness analysis is not fully described, firstly in the terms of importance criteria for nodes under targeted attack.
- Since comments on targeted attacks simulations results are influenced by the degree distribution, it needs to be discussed in the result section.
Some other minor issues:
- Line 86, the reference number is missing. For example, you can cite the classic paper Albert, R., Jeong, H. and Barabási, A-L. (2000). Error and attack tolerance of complex networks, Nature, Vol. 406, pp.378–382, or a more recent Ferraro, G., Iovanella, A., (2018). Clairvoyant targeted attack on complex networks, International Journal of Computational Economics and Econometrics, 8 (1), pp. 41-62.
- Text in Figures 2 and 4 is too small.
- Line 269, 276 and 283, please correct the typos on the capital flow, initial demand and supply chain flow variables.
Round 2
Reviewer 2 Report
The authors did a remarkable job in improving the paper and I am happy to list just a few minor comments that should be considered before the publication of the manuscript.
In particular:
- Formula (1) on page 9 is never introduced.
- Line 365: The definition of weighted degree is not clear. If only the number of neighbours is considered, then in Social Network Analysis such measure is called “Degree”, while the same theory uses the term “Strength” when all the link weights of the neighbours are summed up.
Author Response
Thanks for your comments. We have introduced the formula(1) and changed the term "Strength" instead of "Weighted degree".